# Socioeconomic Impacts of University–Industry Collaborations–A Systematic Review and Conceptual Model

**João Cardim Ferreira Lima** [1,*], **Ana Lúcia Vitale Torkomian** [1], **Susana Carla Farias Pereira** [2], **Pedro Carlos Oprime** [1] **and Luciana Harumi Hashiba** [2]

1   Department of Production Engineering, Federal University of São Carlos (UFSCar),
    São Carlos 13565-905, Brazil; torkomia@ufscar.br (A.L.V.T.); pedro@dep.ufscar.br (P.C.O.)
2   Department of Operations Management, Getúlio Vargas Foundation (FGV EAESP),
    São Paulo 01313-902, Brazil; susana.pereira@fgv.br (S.C.F.P.); luciana.hashiba@fgv.br (L.H.H.)
*   Correspondence: jcflima@estudante.ufscar.br

**Abstract:** University–industry collaborations create socioeconomic impacts for the areas where they are undertaken. Although these collaborations have recognized importance and a high potential to generate economic and social benefits, there is no consensus in the literature on a consolidated conceptual model for assessing their socioeconomic impacts. Given this scenario, this study reviews 94 studies on the socioeconomic impact of university–industry collaborations using a context–intervention–mechanism–outcomes configuration. The impacts identified in the systematic literature review are classified into: (1) economic, (2) social, and (3) financial. The systematic literature review also indicates that the impact of collaborations can change the context and enhance the mechanisms of technology transfer. From a theoretical viewpoint, this work contributes to the structuring of a conceptual model for assessing the socioeconomic impacts of university–industry collaborations. In addition, the results have contributions for management in each strand of the triple helix: they may be useful to guide universities and companies on how to assess the socioeconomic impacts of each collaboration, direct public agents in the evaluation of results of investments, and support the development of policies for innovation and technology management.

**Keywords:** university–industry; economic development; innovation; socioeconomic development

## 1. Introduction

Firms must continually adjust and change to thrive in a competitive, globalized economy. Despite the constant shift, firms drive markets by exploiting and strategically managing knowledge. Markets are driven by creative, efficient, and strategic knowledge management. Universities using knowledge to generate competitive advantage makes them fundamental elements in the science, technology, and innovation ecosystems [1].

The open innovation paradigm points out that firms must carry internal and external knowledge management in order to enhance the internal innovation process of companies, making it faster through the application of both internal and external ideas, with the improvement of its technology [2].

The university is a valuable resource in the open innovation dynamics, as well as a great source of ideas for companies. In addition, academic specialists are trained and have the required resources for technical feasibility evaluation of new technologies implementation. Thus, for the open innovation study area, it is extremely strategic, the analysis and understanding of the socioeconomic impacts of university–industry collaborations.

The triple helix thesis proposes that universities are increasingly vital to discontinuous innovation in knowledge-based societies, superseding companies as the primary source of future economic and social development. The three members of the triple helix are these: industry (as the locus of production); government (as the source of contractual

ties that ensure secure interactions and exchange); and universities (as the source of new information and technology, the generative concept of knowledge-based economies) [3].

In the innovative university–industry–government triple helix, three institutional spheres interact to achieve innovation. Any one of them can take the lead as the organizer of innovation. The broad goals of the three actors are uniform: they all strive for innovation, even they follow different strategies to achieve that goal. Thus, the university–industry–government triple helix is in alignment [4]. There has been a growing recognition of the triple helix's potential contribution to economic development, especially in the relationship between universities and companies [5].

Entrepreneurial ecosystems, organized environments that promote the success of new ventures, come in many forms, including academia [6]. Entrepreneurial universities play critical roles in various triple helix configurations, jump-starting regional innovation by creating a new academic function, economic development [5,7].

The general theory of the economics of entrepreneurial ecosystems differs from the traditional neoclassical theory of economics. Entrepreneurial ecosystems are multifirm and multiproduct markets that might exist in the future; the traditional neoclassical theory of economics cannot capture the combinations of multifirm and multiproduct markets [6].

The metrics to measure the successes and impacts of technology transfer outputs have not yet been well defined [8]. There are several ways universities can positively impact local economies' development beyond technology transfer. However, university-led knowledge-based economic development needs time and patience, which are not always in sync with political schedules [9].

Despite the incentives and an increasing commitment to developing entrepreneurship practices at universities, better information management is still needed, including tools to analyze the entrepreneurial activities' performance. We need broader analysis methods for university entrepreneurship that go beyond specific indicators (e.g., financial returns on intellectual property) and consider the broader social and economic benefits (e.g., knowledge dissemination, creation of intangible assets, employment, socioeconomic and cultural development) [10]. We must develop better metrics to measure the impact and performance of technology transfer [8]. The effectiveness of technology transfer activities can be expressed through such parameters as the social impact on the community, job creation, and poverty reduction, which are all associated with long-term financial benefits [8]. Most university–company collaborative research focuses on specific elements, resulting in fragmented and inadequate research [5].

Consequently, this study sought to provide an embracing understanding of the socioeconomic impact of university–industry collaborations through a systematic literature review; the review addresses the context in which these interactions occur, the mechanisms or channels for technology transfer, and the resulting socioeconomic impacts. The systematic literature review reveals several lines of thought. This article is structured as follows. Section 2 describes the research method, followed by a presentation of the results in Section 3. Section 4 refers to a discussion on the socioeconomic impacts found in literature, the developed conceptual model, and future research directions. Section 5 concludes the article.

## 2. Materials and Methods

The systematic literature review has been widely used in management research as a research strategy aimed at situating the literature on a given topic in a systematic, transparent, and replicable manner [11,12]. A rigorous literature review should follow a well-defined method that provides detailed explanations of how it was conducted and the relevant works selected so others could reproduce the review following the same steps. Systematic literature reviews analyze and synthesize the works published by researchers and academics [13]. Tranfield et al. [12] propose a systematic literature review framework based on a three-step approach to provide evidence-informed management knowledge: (1)

review planning, (2) review conducting, and (3) results reporting [13]. Figure 1 summarizes these stages.

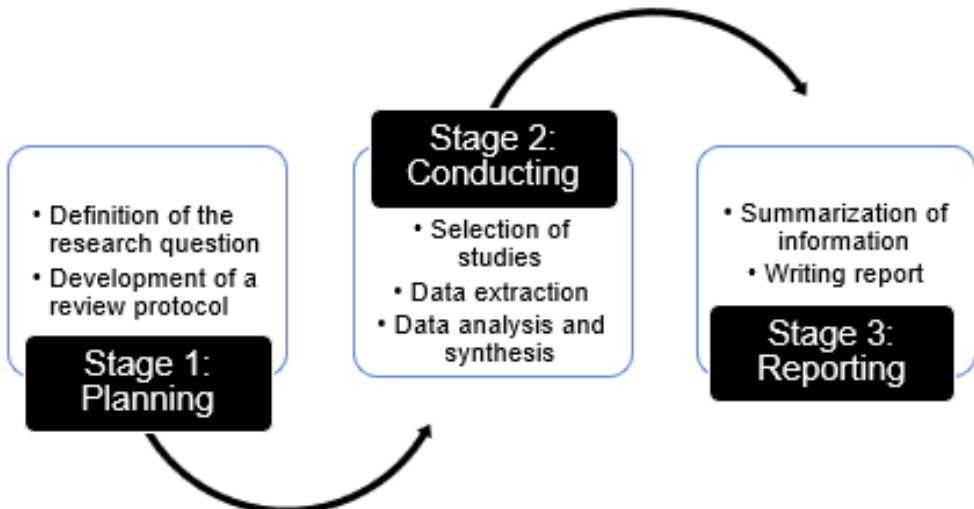

**Figure 1.** Systematic literature review stages.

### 2.1. CIMO Analysis

CIMO analysis is suitable for research that seeks to generate prescriptive knowledge. The CIMO-logic incorporates certain types of interventions to generate mechanisms that achieve the intended results [14]. CIMO helps explain the socioeconomic impacts of university–industry collaborations because it contextualizes the collaborations and inter-actions (i.e., interventions) between universities and businesses that affect both parties' activities and the mechanisms that generate the socioeconomic impacts.

Design proposals in traditional management literature often adopt simplistic Input-Outcome-logic [14], ignoring the outcomes' context-dependence or the mechanisms that produce the outcomes. In practice, concept proposals based on CIMO logic and derived from academic research often include an extensive learning process rather than the direct application of basic rules [14].

### 2.2. Planning (Stage 1)

In the planning of the systematic literature review, the research question and the keywords used were defined and a review protocol elaborated, which is described in Table 1.

First, we defined the research question and chose the keywords. We searched two databases, Scopus and Web of Science, considered the most relevant databases with the largest number of studies on the topic of interest. We then defined the inclusion and exclusion criteria according to studies on systematic literature reviews and research objectives. The search is described in Section 2.2. The topics evaluated in the data extraction are presented in Table 2. For our analysis and synthesis, we used two computer programs: (1) we used the StArt[1] software to select articles by evaluating the titles, abstracts, and keywords based on the inclusion and exclusion criteria (we conducted a peer review of the selected studies to remove inappropriate documents); and (2) we used the NVivo[®2] software for data extraction, data management, and content analysis of the studied theme.

**Table 1.** Review protocol.

| Steps | Description |
|---|---|
| Research question definition | What are the direct socioeconomic impacts of university–industry collaborations? |
| Keywords definition | ("knowledge transfer" or "technology transfer" or "collaboration" or "contract" or "interaction") and ("university" or "academic" or "faculty") and ("social" or "economic" or "socioeconomic") and ("growth" or "impact" or "effect" or "development" or "spillover" or "progress" or "sustainability") and ("firm" or "business" or "company" or "industry" or "corporation" or "establishment" or "organization" or "enterprise") |
| Definition of databases | Web of Science and Scopus |
| Articles' inclusion/exclusion criteria | Main selection criterion: adherence of the article to the topic "University–industry collaboration"<br>• Checking the title, abstract, and keywords, and if necessary, a more in-depth reading of the text<br>• Inclusion criteria: only articles mentioning the socioeconomic impacts of the university–industry collaboration and that present resulting impacts<br>• Articles in English<br>• Articles published in peer-reviewed journals<br>• Main exclusion criterion: Articles published in congress proceedings, book chapters, theses or dissertations, newspaper reports, opinion pieces, and other similar papers<br>• Exclusion of duplicated articles<br>• Exclusion of articles that are studies in progress/unfinished<br>• Exclusion of articles that do not meet the other inclusion criteria |
| Data extraction | • Authors<br>• Journal<br>• Country<br>• Year<br>• Nature of the study (if conceptual or empirical research, if it follows a qualitative or quantitative method or analysis)<br>• Socioeconomic impact |
| Analysis and synthesis | • Use of StArt for the selection of articles in the systematic review<br>• Use of NVivo for data management and content analysis<br>• Data summary: Descriptive statistics and content analysis<br>• CIMO analysis |

Notes: CIMO = context–intervention–mechanisms–outcome [14].

**Table 2.** Number and percentage of articles per journal.

| Journal | Number | Percentage (%) |
|---|---|---|
| Journal of Technology Transfer | 18 | 19.15 |
| Research Policy | 9 | 9.57 |
| Technological Forecasting & Social Change | 8 | 8.51 |
| Science and Public Policy | 5 | 5.32 |
| Economic Development Quartely | 5 | 5.32 |
| R&D Management | 3 | 3.19 |
| Entrepreneurship & Regional Development: An International Journal | 2 | 2.13 |
| Growth and Change | 2 | 2.13 |
| Higher Education | 2 | 2.13 |
| Industry & Higher Education | 2 | 2.13 |
| Journal of the Knowledge Economy | 2 | 2.13 |
| Management Decision | 2 | 2.13 |
| Science, Technology & Society | 2 | 2.13 |
| Technovation | 2 | 2.13 |
| Economy of Region | 1 | 1.06 |

**Table 2.** *Cont.*

| Journal | Number | Percentage (%) |
|---|---|---|
| Economic Research | 1 | 1.06 |
| Engineering Economics | 1 | 1.06 |
| Evaluation and Program Planning | 1 | 1.06 |
| Foresight | 1 | 1.06 |
| Futures | 1 | 1.06 |
| Industrial and Corporate Change | 1 | 1.06 |
| Innovation | 1 | 1.06 |
| Innovative Higher Education | 1 | 1.06 |
| International Entrepreneurship and Management Journal | 1 | 1.06 |
| International Journal of Entrepreneurial Behavior & Research | 1 | 1.06 |
| International Journal of Global Environmental Issues | 1 | 1.06 |
| International Journal of Sustainability in Higher Education | 1 | 1.06 |
| International Journal of Technology Management | 1 | 1.06 |
| Journal of Business Research | 1 | 1.06 |
| Journal of Business Venturing | 1 | 1.06 |
| Journal of Chinese Economic and Business Studies | 1 | 1.06 |
| Journal of Intellectual Capital | 1 | 1.06 |
| Journal of Product Innovation Management | 1 | 1.06 |
| Journal of Regional Science | 1 | 1.06 |
| Knowledge Management Research and Practice | 1 | 1.06 |
| Measuring Business Excellence | 1 | 1.06 |
| Prometheus: Critical Studies in Innovation | 1 | 1.06 |
| Regional Studies | 1 | 1.06 |
| Research Evaluation | 1 | 1.06 |
| Social Science | 1 | 1.06 |
| Social Science Information | 1 | 1.06 |
| Studies in Regional Science | 1 | 1.06 |
| Tertiary Education and Management | 1 | 1.06 |
| The Annals of Regional Science | 1 | 1.06 |
| Total | 94 | 100 |

### 2.3. Conducting (Stage 2)

We conducted our keyword search in June 2020. Our search for relevant articles published between 1945 and 2019 turned up 2516 articles: 1488 (59%) listed by Web of Science and 1028 (41%) listed by Scopus. We imported the research data from the databases into the StArt software in BibTeX format. Duplicate articles (393) were removed, leaving a total of 2123 articles. We read the titles, abstracts, and keywords and applied the inclusion and exclusion criteria identified in Table 1, which left us with 180 articles. After evaluating the full texts based on the above-mentioned criteria, we retained 94 articles for the study (86 did not qualify for inclusion). Figure 2 shows the literature filtration process.

We extracted the following information: authors, year of publication, title, journal, nature of the study (conceptual or empirical), methodology [15–17], and the country of origin of the author's institution [16,17]. We used the context–intervention–mechanisms–outcome (CIMO) methodology to conduct our analysis [14]. The CIMO analysis considers a context in which an intervention is suggested, creating mechanisms that, in a certain context, are triggered by the intervention to achieve the intended outcome(s) [14]. In the context of this study, CIMO refers to how university–industry collaborations are carried out, considering the context in which they occur, as well as interventions, mechanisms, and results in terms of socioeconomic impacts.

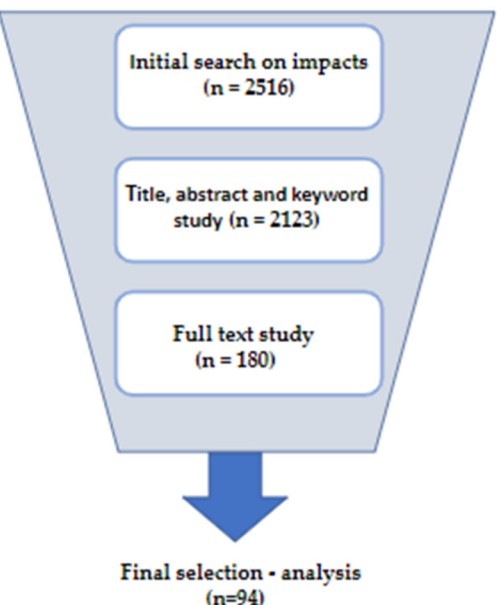

**Figure 2.** Selection of studies.

## 3. Results

This section shows the results (Stage 3) with the descriptive analysis.

*Descriptive Analysis*

Table 2 illustrates the production of scientific articles and classifies them based on the journals with the highest publication volume. The selected papers have been published in a variety of academic journals. The articles, numbers, and percentages of publication in each journal are shown.

The authors who participated in more than one article were David Urbano (4), Albert Link, Christopher Hayter, David Audretsch, Erik Lehmann, Matthias Menter, Maribel Guerrero (3), and Andrés Barge-Gil, Aurelia Mondrego, Helen Lawton Smith, Peter Nijkamp, Joaquín Azagra-Caro, Elena Tur, Magnus Klofsten, Alain Fayolle (2) (Appendix A). The remaining authors contributed to only one article each.

Figure 3 illustrates the overall increase in the number of articles published on the subject during the selected period. Figure 3 shows a trend of significant growth in publications since 2010, with the highest number of published articles in 2019, with 17. In 2019, the *Journal of Technology Transfer* published the special issue "Economic, Technological and Societal Impacts of Entrepreneurial Ecosystems" and the journal *Technological Forecasting & Social Change* published the special issue "Understanding Smart Cities: Innovation Ecosystems, Technological Advancements, and Societal Challenges."

Our analysis of the countries that produced scientific articles considered the countries of the institutions to which the authors and co-authors were linked. If an author was linked to more than one institution in different countries, we considered all the institutions. The greatest percentage of researchers were linked to institutions in the United States (25 articles; 27%), the United Kingdom (15 articles; 16%), and Spain (12 articles; 13%) (Appendix A). All 94 articles addressed socioeconomic impacts of university–industry collaborations and presented relevant information for the construction of the conceptual-theoretical framework of this research. Each article analyzed university–industry collaborations under a specific perspective on a main theme that was identified in the theoretical construction of each study.

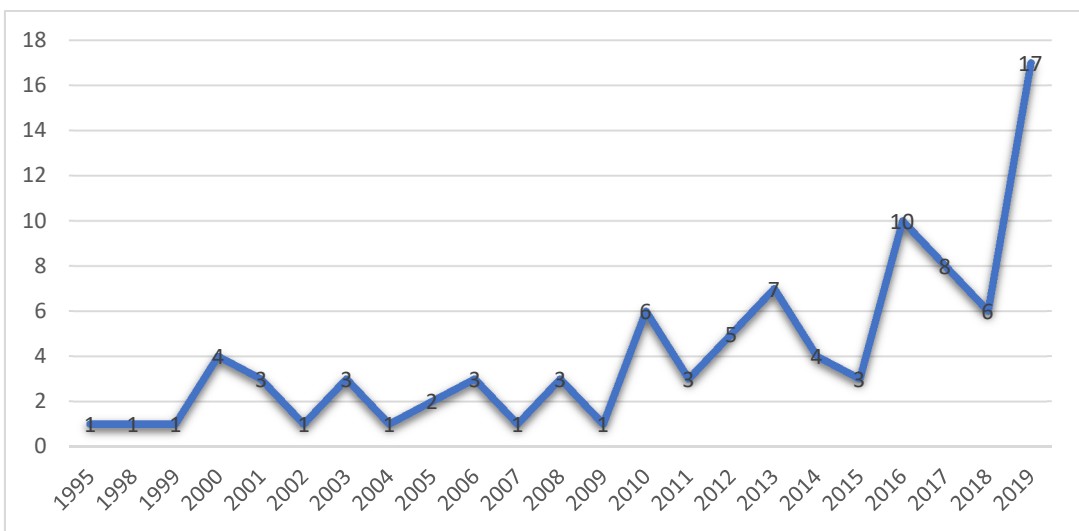

**Figure 3.** Frequency of publications (1995–2019).

This literature review divided the 94 articles into two methodological categories: conceptual and empirical. Conceptual studies are those that formulate emerging concepts, frameworks, and models. Empirical studies are those that use surveys, case studies, interviews, and experiments [17,18]. Our evaluation found more empirical studies (85) than conceptual studies (9). In the empirical research, the predominant methods were surveys (57), case studies (21), interviews (6), and experiments (1), as shown in Table 3.

**Table 3.** Research methods.

| Research Method | Quantity | Percentage (%) |
| --- | --- | --- |
| Conceptual | 9 | 9.57 |
| Empirical | 85 | 90.43 |
| Survey | 57 | 60.64 |
| Case Study | 21 | 22.34 |
| Interview | 6 | 6.38 |
| Experiment | 1 | 1.06 |

## 4. Discussion

This section presents the discussion based on the CIMO structure. Section 4.1 presents the CIMO analysis and discusses the context of university–industry collaborations, the intervention, the results, and the mechanisms that lead to the results.

### 4.1. Context

In the CIMO perspective, the contexts analyzed are the internal and external environments that influence behavioral change [14,16]. This systematic literature review identified both external and internal contexts: (1) the external contexts were the socioeconomic conditions and the national and regional laws and policies; (2) the internal contexts were the universities' characteristics, the firms' characteristics, and the researchers' characteristics. A region's capacity to absorb knowledge is most often associated with its socioeconomic conditions [19]. The ability of universities to invest in research and development (R&D to generate knowledge and apply it in industries generating innovations depends on political, economic, and social conditions [9].

Technology transfer policies support a commitment that considers knowledge spillovers to be public and offers property rights to guarantee the commercialization of developed technologies and a return on additional private investments. In the innovation system, the political and legal environment influences the type of knowledge generated, prioritizing



the areas of greatest interest and directing investments, affecting the rate of technological transformation [9]. Therefore, consolidating entrepreneurial universities created national and regional programs and public policies to encourage university–industry collaborations; this benefited local companies and opened a new market for academic innovation R&D [20].

Universities and companies follow distinct paradigms and have different interests and objectives, the latter totally focused on profits and financial returns, and the former with their own interests. However, universities are under increasing pressure to generate economic benefits for society [19]. Universities invest financial and intellectual capital in startups in exchange for part of the businesses created from scientific research. They also establish collaborations with technology companies, based on R&D in exchange for participation in the generated intellectual property and benefits to the status of their faculty [21].

Commercial companies have the same relatively simple goal: earning profits. In contrast, universities have multiple objectives beyond the obvious ones of educating students; they also serve the greater society by developing and sharing knowledge and nurturing their faculty, scientists, and researchers to support the scientific community in general [9]. Research in collaborations between universities and industries should focus on areas of mutual interest, both academic and business. For a collaboration to be sustainable, the research results must add long-term value for the university and the industry or company. The value will depend on the perceptions of the research's impact on enhancing companies' and universities' strengths [22].

Several authors have reported on how various firm characteristics influence the establishment of university collaborations: size [22,23]; time of existence [24]; geographic location [21,25,26]; operating sector [19]; and specialization in the operating sector [27]. Ahrweiler et al. [28] investigated the role of university–industry links for innovation generation and diffusion in networks in two contexts: large, diversified companies and small technology companies. The latter context has been studied by several authors, such as Audretsch et al. [29] and Doh and Kim [24].

Although favorable external contexts (socioeconomic conditions, national and regional laws and policies) and favorable internal contexts (companies' and universities' characteristics) are necessary, they are not sufficient to ensure technology transfer. Furthermore, although cutting-edge research universities are critical assets for urban and regional economies, their presence does not guarantee regional economic development [25].

Ahrweiler et al. [28] found no direct and instant link between increasing knowledge inputs and financial returns with increasing profitability; nor did they find that companies with collaborative projects with universities were any better at adapting to changes in environmental conditions than their nonaffiliated counterparts. The average life of companies that interacted with universities was no longer than that of those that did not; additionally, increasing the knowledge quantity input automatically did not elevate the innovation generated or economic benefits.

The context presented by Bramwell and Wolfe [25], Bercovitz and Feldman [9], and Ahrweiler et al. [28] showed that despite the existence of robust structures with favorable conditions for the transfer of technology and the establishment of university–industry collaborations, the objectives of the collaborations were not always realized. This evidences the need for and importance of another factor in collaborations: the people and personal characteristics critical for technology transfer. The participants must connect academic research and its industrial and marketing applications, transforming scientific knowledge into financial profit. Effectively managing the available resources is essential for competitive advantage. Researchers and those involved in collaborations with access to cutting-edge technological research must identify the opportunities for pioneering innovations in the market efficiently and competitively.

Bradley et al. [30] outlined the various challenges for technology transfer: (1) university entrepreneurs are often older and generally lack many relevant business skills; (2) product

research faculties are not always willing to adapt or align their research to technologies that can be transferred; (3) universities often lack the strong and consolidated social network necessary for successful technology transfer; and (4) university policies (e.g., promotion and tenure, financial and intellectual property) often do not offer the necessary subsidies and motivations for faculties to participate in technology transfer activities.

Figure 4 shows the context of university–industry collaborations.

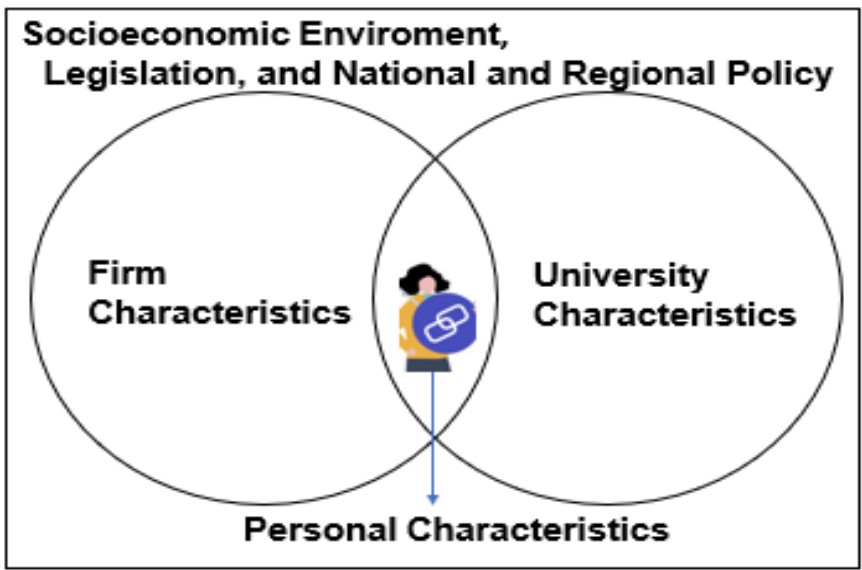

**Figure 4.** Context of university–industry collaborations.

*4.2. Interventions*

Interventions are inserted in a broader system, the social system [18]. They are influenced by interpersonal links, the institutional configuration, and the broadest infrastructural system [14,31]. Managers have interventions at their disposal to influence behavior [18].

University–industry interactions are multifaceted, complex, and diverse. Commercialization can include a wide variety of transactions between universities and industries [9]. Although the flow of knowledge drives innovation, knowledge transfer from university to company is fluid, complex, and iterative [25]. Many authors have found formal and informal links in university–industry interactions: Budyldina [20], Bercovitz and Feldman [9], Bramwell and Wolfe [25], Ahrweiler et al. [28], Dutrénit and Arza [32], Perkmann et al. [33], Hope [34], Lendel and Qian [35], Azagra-Caro et al. [36], Kochetkov et al. [37], and Owusu-Agyeman and Fourie-Malherbe [38].

Numerous formal and informal empirical works have investigated the possible ties between universities and firms. Universities are expected to provide the permanent growth, development, and diversification of knowledge for potential transfer to the industry that drives innovation. Furthermore, universities are strong network partners that are considered highly reliable because they are tied to public investments that largely isolate them from market fluctuations [28].

University–industry collaborations associate formal and informal interactions and are affected by industries' characteristics and business strategies, universities' rules, and the operational mode of the technology transfer activities and government policy interests [9]. The interactions between universities and industries frequently start as informal relationships that develop into more formal relationships with detailed descriptions of planning, roles, and expectations [38]. Formal channels involve the contractually supervised use of universities' and firms' skills, resources, and facilities. In the absence of a formal contract, informal channels provide access to a pool of knowledge reflected in skills, resources,

technological and scientific capacities and requirements, and the preparation, procurement, and distribution of skilled personnel [36].

Commercialization generally occurs outside of formal academic channels, and universities seldom keep track of it [33]. Local economic effects are generally the result of a complex, dynamic, temporally unfolding series of interactions between formal and informal channels of knowledge transfer [36]. Knowledge created during formal interactions can be transferred through informal networks [36].

### 4.3. Mechanisms

Mechanisms produce outcomes [14]. In the context of university–industry collaborations, the mechanisms are the channels for technology transfer. We analyzed the links between contexts, interventions, and outcomes to establish the mechanisms. Table 4 shows the results by computing the percentage of each dominant mechanism. Appendix B shows the citations for each article used in the CIMO analysis, obtained from Google Scholar (5 April 2021), including the authors, year of publication, and the dominant mechanism.

**Table 4.** Dominant mechanism.

| Dominant Mechanism | Presences | Percentage % |
|---|---|---|
| Intellectual property | 18 | 19.15 |
| Spin-offs | 15 | 15.96 |
| Spin-offs and intellectual property | 15 | 15.96 |
| Hybrid organizations | 13 | 13.83 |
| Sponsored research | 11 | 11.70 |
| All mechanisms | 10 | 10.64 |
| Consulting and hiring professionals with academic knowledge | 7 | 7.47 |
| Spin-offs and hybrid organizations | 2 | 2.13 |
| Spin-offs and sponsored research | 1 | 1.06 |
| Intellectual property and hybrid organizations | 1 | 1.06 |
| Intellectual property and publications | 1 | 1.06 |
| Total | 94 | 100% |

The mechanisms identified were intellectual property, spin-offs, hybrid organizations, sponsored research, consulting and hiring professionals with academic knowledge, and publications and conferences. Table 4 shows the dominant mechanisms. Intellectual property (47.87%) and spin-offs (45.75%) stood out from the rest of the dominant mechanisms. The relevance of intellectual property has been noted by Perkmann et al. [33], Mets et al. [39], Jones and De Zubielqui [40], and Secundo et al. [41]. Licensing intellectual property provides legal rights that give companies access to technological solutions in the universities' intellectual property [9]. Spinning off companies and hiring professionals with academic knowledge enables more straightforward technology transfers through human resources movement [9]. Chiesa and Piccaluga [42] called academic spin-off enterprises one of the most promising ways to get scientific findings to the market.

The triple helix concerns the relationships among universities, industries, and governments and the creation of such hybrid organizations as incubators, science parks, and technology transfer offices [3]. The original business support structure of incubation has been reconsidered to emphasize its focus on the educational mission in training organizations [3]. According to Guadix et al. [43], considering the regional economic, business, and industrial context, science and technology parks have a high strategic value for the regions where they are located and carry out operations that promote research, development, innovation, and technology transfer. Universities transfer internally developed technologies to the public domain via technology transfer offices [19]. Audretsch et al. [29] emphasized the importance of technology transfer offices in universities' technology licensing. Bercovitz and Feldman [9] maintained that the setting of technology transfer offices represents an independent variable that partially accounts for the evaluated differences in patenting, licensing, and sponsored research between institutions.

Technology transfer offices differ considerably in their commercialization capacity. The license income distribution is highly localized, with a few big commercial hits yielding strong profits for a few universities [9]. Many high-impact start-up projects have emerged from academic studies in many developed countries, with the majority of these firms originating with a limited group of strongly entrepreneurial universities [44].

Sponsored research is a contract between a university and an industry. A sponsored research project supports university-commissioned studies and offers funding for facilities, graduate students, course launches, and faculty summer care [9]. Examples include collaborative research [45,46], contract research [22,35,47–50], and the establishment of R&D organizations [22,51–53].

Several authors considered consulting and hiring professionals with academic knowledge an important mechanism, such as Bramwell and Wolfe [25], Breznitz and Feldman [19], Chen et al. [51], and Hope [34]. Universities do not usually have individual consultancy agreements with the faculty member(s), as companies nearly always own all the created intellectual property and directly remunerate the faculty member; in these cases, the university does not have access to new investments and potential generation of intellectual property [9].

Dutrénit and Arza [32] argued that publications and conferences are traditional technology transfer mechanisms. They classified mechanisms into four types: (1) traditional (hiring professionals with academic knowledge and publication and conferences); (2) services (providing science and technical resources in exchange for funds, such as consulting, use of quality management facilities, tests, instruction, and so on); (3) commercialization of scientific results already obtained (academic spin-offs, licensing, patents, and incubators); and (4) bidirectional mechanisms motivated by long-term aims of knowledge (contract research, joint R&D projects, and scientific–technological parks). Their model was also used by Orozco and Ruiz [54] and Fernandes et al. [55]. Serendipity is considered an unconventional mechanism that could possibly start relationships that later unfold through different mechanisms [9].

University offices are often regarded as displays for companies and treated as cooperation platforms for marketing their R&D results. The mechanisms vary depending on the context in which a university and a company are engaged (e.g., the country, region, and prevailing incentive policies). Hayter and Link [56] listed numerous university-affiliated proof-of-concept centers (PoCCs) in the United States that contributed to a rise in that country's academic spin-offs. Chang et al. [57] presented a model created in China of a university–industry cooperation platform in which companies could seek partnerships with any higher education university in the country or vice versa. The China cooperation platform has improved the economic performance of that country's high-tech companies; this suggests a positive connection between economic performance and the number of cooperating parties. Different cooperation mechanisms impact the economic performance of high-tech companies at different levels [32,57].

### 4.4. Outcomes

In this systematic literature review, the outcomes are the socioeconomic impacts of the university–industry collaborations. We classified the outcomes into three dimensions: (1) economic, (2) social, and (3) financial. We further subdivided each dimension as follows: (1) economic: infrastructure, production and processes, and scientific development; (2) social: jobs, skills, and qualification; and (3) financial: purchases, taxes, investments, and income generation. Figure 5 shows the proposed model for measuring the economic impact of university–industry collaborations.

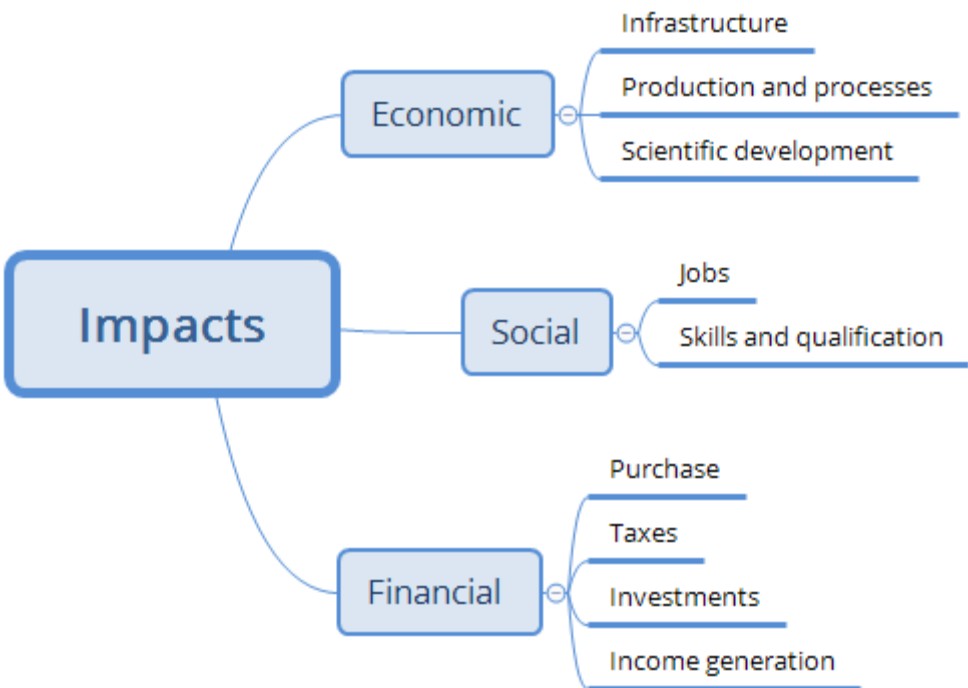

**Figure 5.** Evaluation model for the socioeconomic impact of university–industry collaborations.

Several authors have addressed some of the socioeconomic impacts of university–industry collaborations on the technology transfer mechanism, such as the emerging of companies (startups and spin-offs), patents and licensing, and relevant scientific publications. Ahrweiler et al. [28] and Urbano and Guerrero [50] claimed that these collaborations could lead to new business opportunities. Etzkowitz [21] contended that universities have emerged as leading actors in a society predicated on knowledge owing to their nature as creators of original ideas. University–industry collaborations often result in new scientific and technological development partnerships that generate intellectual properties and market opportunities, such as industrial applications and new enterprises. Scientific novelty is of interest to academics, too, because it can generate new avenues for research. An enhanced mechanism from a university–industry collaboration can directly lead to such positive results as higher productivity, new products, increased sales, and commercial and societal value creation. Most of the authors in the systematic literature review regarded job creation as a socioeconomic impact of university–industry collaborations that could be quantified and influences people's quality of life.

Entrepreneurial universities can contribute through an advisory role in public policy formulation [19,46,58]. In this role, universities engage with local communities on a variety of themes. Nevertheless, most of the services and activities supplied by institutions cannot be easily quantified [19]. A university–industry collaboration can have several socioeconomic impacts on the actors in [59] triple helix; therefore, we propose a conceptual model of socioeconomic impact based on the main benefits from the actors in the triple helix. Figure 6 illustrates our Socioeconomic Triple Helix Conceptual Model.

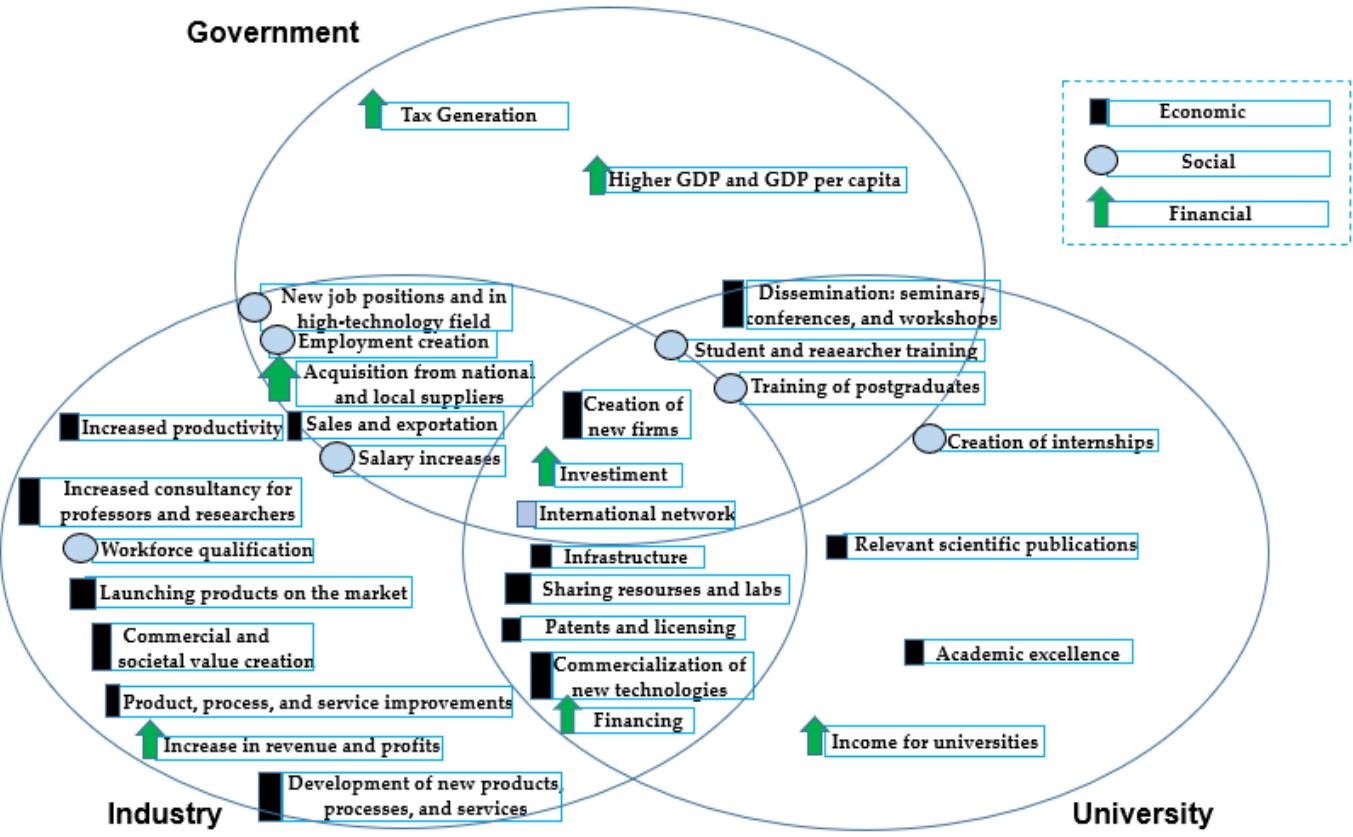

**Figure 6.** Socioeconomic triple helix.

The triple helix model puts the institutional spheres into perspective. An understanding of the most significant impacts and the stakeholders who benefit from such impacts facilitates negotiation between the constituents and enables strategies to be defined with the objective of enhancing the socioeconomic impacts based on interests and priorities.

The advantage of organizing the model according to the triple helix thesis is that the model has a visual and didactic advantage that makes it possible to quickly map the impacts and the main stakeholders, allow cuts or partial indicator applications for more specificity, and evaluate the impact of particular actions or public policies.

## 5. Conclusions

University–industry collaborations can have appropriate economic and social advantages. We developed the socioeconomic triple helix, a conceptual model of socioeconomic impacts identified in the systematic literature review based on Etzkowitz and Leydesdorff's [59] triple helix model. Our model has significant academic and managerial contributions.

### 5.1. Theoretical Contributions

Many authors, including Galan-Muros and Davey [5], Audretsch et al. [6], Alessandrini et al. [8], Bercovitz and Feldman [9], and Etzkowitz et al. [10], have claimed that traditional metrics and indicators cannot capture the socioeconomic benefits of university–industry collaborations. Our work enables a deeper analysis of the socioeconomic impacts of university–industry collaborations, highlighting the existing effects in the literature through synthesizing high-value insights into the theory of socioeconomic development based on strategic knowledge management, R&D, and technological innovation. Our model complements the triple helix model with a socioeconomic perspective of the interactions among government, universities, and industries, thus adding knowledge and

elaborating on the theory. This work provides a guide for researchers and scholars who are interested in university–industry collaborations.

### 5.2. Managerial Contributions

In addition to its academic contributions, this research and our new conceptual model benefit all the actors in the triple helix: (1) universities and companies can use the model to assess the socioeconomic impacts of individual collaborations; (2) public agents can use it to evaluate the impacts of their investments; and (3) government agencies can use it to inform their development of public policies for innovation and technology management. The CIMO analysis enabled us to arrive at a deeper understanding of the peculiarities of university–industry collaborations and the generated socioeconomic results. CIMO made it possible to modify the contexts in which collaborations were undertaken to create a more conducive environment for the knowledge-based socioeconomic development that enables new public policies and mechanisms to enhance technology transfer.

### 5.3. Research Limitations

The limitation of the research is that the model is generic, the types and areas of university–industry collaboration and their specific characteristics for each one must be taken into account in order to understand which indicators have the greatest strategic value in your institution's position. Another important aspect to observe is the phase of university–industry collaboration, applying the most significant, important indicators and with the greatest variations in impact on that phase.

### 5.4. Future Research Directions

Based on the results and the discussion on the socioeconomic impact of university–industry collaborations, we offer a few suggestions for future research: (1) an application of an evaluation model to university and companies and (2) a development of methods for the indirect impact assessment in local communities.

Future research should pursue applications of the proposed model, which will require developing metrics for each indicated variable. These additional metrics will enable the assessment of the socioeconomic impact of collaborative activities of university–industry partnerships by creating indicators that can be controlled and enhanced based on actions focused on the technology transfer mechanisms. Research has shown that conventional and quantitative metrics are not sufficient to measure the socioeconomic impact of university–industry collaborations fully [9,20]. In addition, a more qualitative assessment is suggested that addresses the indirect impact of university–industry collaborations—for instance, the creation of public policies [19,46], regional human capital attraction [5], and community and city development [19].

**Author Contributions:** Abstract and Introduction, J.C.F.L. and A.L.V.T.; Materials and Methods, J.C.F.L., A.L.V.T., S.C.F.P. and L.H.H.; Results J.C.F.L., A.L.V.T. and P.C.O.; Discussion, J.C.F.L., A.L.V.T. and S.C.F.P.; Conclusions, J.C.F.L. and A.L.V.T. All authors have read and agreed to the published version of the manuscript.

**Funding:** This work was only possible thanks to the funding provided by Coordenação de Aperfeiçoamento de Pessoal de Nível Superior (CAPES)—PhD scholarship, number: 88882.426281/2019-01.

**Institutional Review Board Statement:** Not applicable.

**Informed Consent Statement:** Not applicable.

**Data Availability Statement:** All the information is shown in the article.

**Acknowledgments:** The authors would like to thank CAPES–Coordenação de Aperfeiçoamento de Pessoal de Nível Superior, UFSCar–Universidade Federal de São Carlos, FGVin-Innovation Center, FGV–Fundação Getulio Vargas and Editage.

**Conflicts of Interest:** The authors declare no conflict of interest.

## Appendix A

**Table A1.** Authors who participated in more than one article.

| Author | Articles | Year |
|---|---|---|
| David Urbano | 4 | 2013, 2016 (2) and 2019 |
| Albert Link | 3 | 2013 (2) and 2015 |
| Christopher Hayter | 3 | 2013 (2) and 2015 |
| David Audretsch | 3 | 2009, 2013, and 2019 |
| Erik Lehmann | 3 | 2015, 2018, and 2019 |
| Matthias Menter | 3 | 2015, 2018, and 2019 |
| Maribel Guerrero | 3 | 2013, 2016, and 2019 |
| Henry Etzkowitz | 2 | 2005 and 2013 |
| Maryann Feldman | 2 | 2006 and 2012 |
| Iryna Lendel | 2 | 2010 and 2016 |
| Andrés Barge-Gil | 2 | 2010 and 2011 |
| Aurelia Mondrego | 2 | 2010 and 2011 |
| Helen Lawton Smith | 2 | 2003 and 2012 |
| Peter Nijkamp | 2 | 2007 and 2014 |
| Joaquín Azagra-Caro | 2 | 2017 and 2019 |
| Elena Tur | 2 | 2017 and 2019 |
| Magnus Klofsten | 2 | 1999 and 2019 |
| Alain Fayolle | 2 | 2016 and 2019 |

**Table A2.** Participation of analyzed countries in the number of publications.

| Country | Participation (Number of Publications) | Percentage (%) |
|---|---|---|
| United States | 25 | 26.60 |
| United Kingdom | 15 | 15.96 |
| Spain | 12 | 12.77 |
| Germany | 8 | 8.51 |
| Italy | 8 | 8.51 |
| The Netherlands | 8 | 8.51 |
| Sweden | 5 | 5.32 |
| Norway | 4 | 4.26 |
| Australia | 3 | 3.19 |
| Canada | 3 | 3.19 |
| Portugal | 3 | 3.19 |
| South Korea | 3 | 3.19 |
| Austria | 2 | 2.13 |
| Belgium | 2 | 2.13 |
| Brazil | 2 | 2.13 |
| Denmark | 2 | 2.13 |
| Mexico | 2 | 2.13 |
| Russia | 2 | 2.13 |
| South Africa | 2 | 2.13 |
| Spain | 2 | 2.13 |
| Taiwan | 2 | 2.13 |
| Argentina | 1 | 1.06 |
| China | 1 | 1.06 |
| Colombia | 1 | 1.06 |
| Costa Rica | 1 | 1.06 |
| Croatia | 1 | 1.06 |
| Czech Republic | 1 | 1.06 |
| Estonia | 1 | 1.06 |
| France | 1 | 1.06 |
| India | 1 | 1.06 |

**Table A2.** *Cont.*

| Country | Participation (Number of Publications) | Percentage (%) |
|---|---|---|
| Indonesia | 1 | 1.06 |
| Ireland | 1 | 1.06 |
| Japan | 1 | 1.06 |
| Lithuania | 1 | 1.06 |
| Malaysia | 1 | 1.06 |
| Nigeria | 1 | 1.06 |
| Singapore | 1 | 1.06 |
| South Africa | 1 | 1.06 |

## Appendix B

**Table A3.** Number of citations and dominating mechanisms.

| Authors and Year | Citations | Dominating Mechanisms |
|---|---|---|
| Ahrweiler et al. (2011) [28] | 189 | Spin-offs |
| Alessandrini et al. (2013) [8] | 27 | Intellectual property |
| Aparicio et al. (2016) [60] | 40 | Consulting and hiring professionals with academic knowledge |
| Audretsch et al. (2013) [29] | 37 | Intellectual property |
| Audretsch et al. (2019) [6] | 131 | Spin-offs and intellectual property |
| Azagra-Caro et al. (2017) [36] | 81 | Intellectual property |
| Azagra-Caro et al. (2019) [61] | 9 | Sponsored research |
| Barge-gil and Modrego (2011) [23] | 107 | Hybrid organizations |
| Baskaran et al. (2019) [62] | 4 | Hybrid organizations |
| Bercovitz and Feldman (2006) [9] | 1070 | All mechanisms |
| Bradley et al. (2013) [30] | 72 | Hybrid organizations |
| Bramwell and Wolfe (2008) [25] | 771 | Intellectual property |
| Breznitz and Feldman (2012) [19] | 267 | Intellectual property |
| Budyldina (2018) [20] | 54 | Spin-offs and intellectual property |
| Carayannis et al. (2017) [63] | 54 | Consulting and hiring professionals with academic knowledge |
| Carlsson et al. (2009) [64] | 276 | Spin-offs and intellectual property |
| Chang et al. (2006) [57] | 62 | Spin-offs and intellectual property |
| Chen et al. (2016) [51] | 13 | Sponsored research |
| Cheshire and Magrini (2000) [65] | 233 | Consulting and hiring professionals with academic knowledge |
| Chiesa and Piccaluga (2000) [42] | 546 | Spin-offs |
| Civera et al. (2019) [66] | 18 | Spin-offs |
| Coronado et al. (2017) [27] | 4 | Intellectual property |
| Dalmarco et al. (2018) [67] | 65 | Spin-offs and intellectual property |
| Dill (1995) [68] | 202 | All mechanisms |
| Doh and Kim (2014) [24] | 285 | Intellectual property |
| Dutrénit and Arza (2010) [32] | 121 | All mechanisms |
| Etzkowitz (2013) [21] | 278 | All mechanisms |
| Etzkowitz et al. (2005) [3] | 490 | Hybrid organizations |
| Fadeyi et al. (2019) [69] | 4 | Spin-offs and intellectual property |
| Farinha et al. (2016) [70] | 92 | Sponsored research |
| Fernandes et al. (2010) [55] | 143 | All mechanisms |
| Fischer et al. (2018) [71] | 10 | Hybrid organizations and intellectual property |
| Galan-Muros and Davey (2019) [5] | 66 | Sponsored research |
| Gjelsvik (2018) [72] | 10 | Spin-offs and intellectual property |
| Guadix et al. (2016) [43] | 63 | Hybrid organizations |
| Guerrero et al. (2016) [73] | 174 | Spin-offs and intellectual property |
| Handoko et al. (2014) [74] | 46 | Consulting and hiring professionals with academic knowledge |
| Hayter (2013) [75] | 112 | Spin-offs |
| Hayter and Link (2015) [56] | 48 | Hybrid organizations |
| Hearn et al. (2004) [76] | 54 | Spin-offs and intellectual property |

**Table A3.** *Cont.*

| Authors and Year | Citations | Dominating Mechanisms |
|---|---|---|
| Holley and Harris (2018) [52] | 15 | Spin-off and sponsored research |
| Hooi and Wang (2019) [77] | 2 | Spin-offs and intellectual property |
| Hope (2016) [34] | 16 | Intellectual property and publications |
| Iacobucci and Micozzi (2015) [78] | 90 | Spin-offs |
| Jones and De Zubielqui (2017) [40] | 61 | Consulting and hiring professionals with academic knowledge |
| Jones-Evans et al. (1999) [79] | 257 | Hybrid organizations |
| Kalantaridis (2019) [80] | 3 | Intellectual property |
| Kim et al. (2012) [81] | 151 | Spin-offs |
| Klofsten et al. (2019) [82] | 101 | Hybrid organizations |
| Kochetkov et al. (2017) [37] | 15 | Spin-offs |
| Kourtit et al. (2014) [53] | 2 | Spin-offs and hybrid organizations |
| Langford et al. (2006) [83] | 173 | Sponsored research |
| Lee (2019) [84] | 7 | Intellectual property |
| Lehmann and Menter (2016) [85] | 60 | All mechanisms |
| Lehmann and Menter (2017) [86] | 40 | Intellectual property |
| Lendel (2010) [26] | 104 | Spin-offs |
| Lendel and Qian (2017) [35] | 8 | Consulting and hiring professionals with academic knowledge |
| Lin (2019) [87] | 2 | Intellectual property |
| Looy et al. (2003) [88] | 203 | Spin-offs |
| Macpherson and Ziolkowski (2005) [48] | 37 | Hybrid organizations |
| Mariani, Carlesi and Scarfò (2018) [89] | 21 | Spin-offs |
| Mascarenhas et al. (2019) [90] | 7 | Intellectual property |
| Mccullough (2003) [91] | 7 | All mechanisms |
| Mets et al. (2016) [39] | 9 | Intellectual property |
| Ndonzuau et al. (2002) [92] | 665 | Spin-offs |
| Núñez-sánchez et al. (2012) [93] | 25 | Sponsored research |
| O'shea et al. (2008) [44] | 574 | Spin-offs |
| Olmos-peñuela et al. (2014) [47] | 142 | Consulting and hiring professionals with academic knowledge |
| Onken et al. (2019) [94] | 1 | Intellectual property |
| Orozco and Ruiz (2010) [54] | 35 | All mechanisms |
| Owusu-Agyeman and Fourie-Malherbe (2019) [38] | 2 | Sponsored research |
| Perkmann et al. (2015) [33] | 62 | Intellectual property |
| Philbin (2008) [22] | 52 | All mechanisms |
| Piirainen et al. (2016) [45] | 39 | All mechanisms |
| Raguž and Mehičić (2017) [95] | 3 | Intellectual property |
| Ramos-Vielba and Fernández-Esquinas (2012) [46] | 93 | Spin-offs and intellectual property |
| Ratinho and Henriques (2010) [96] | 413 | Hybrid organizations |
| Roessner et al. (2013) [97] | 96 | Intellectual property |
| Sá et al. (2019) [98] | 27 | Sponsored research |
| Sánchez-Barrioluengo and Benneworth (2019) [99] | 61 | Sponsored research |
| Secundo et al. (2017) [41] | 115 | Spin-offs and intellectual property |
| Sherman and Chappell (1998) [100] | 267 | Hybrid organizations |
| Sizer (2001) [101] | 58 | Spin-offs and intellectual property |
| Smith (2003) [102] | 145 | Hybrid organizations |
| Smith and Bagchi-Sem (2012) [58] | 157 | Spin-offs and intellectual property |
| Steffensen et al. (2000) [49] | 700 | Spin-offs |
| Urbano and Guerrero (2013) [50] | 297 | Spin-offs |
| Van Geenhuizen et al. (2007) [103] | 4 | Hybrid organizations and spin-offs |
| Varga (2000) [104] | 662 | Spin-offs and intellectual property |
| Villasana (2011) [105] | 27 | Hybrid organizations |
| Vincett (2010) [106] | 139 | Spin-offs |
| Wakkee et al. (2019) [107] | 31 | Sponsored research |
| Wen and Kobayashi (2001) [108] | 91 | Sponsored research |
| Zucker and Darby (2001) [109] | 443 | Intellectual property |

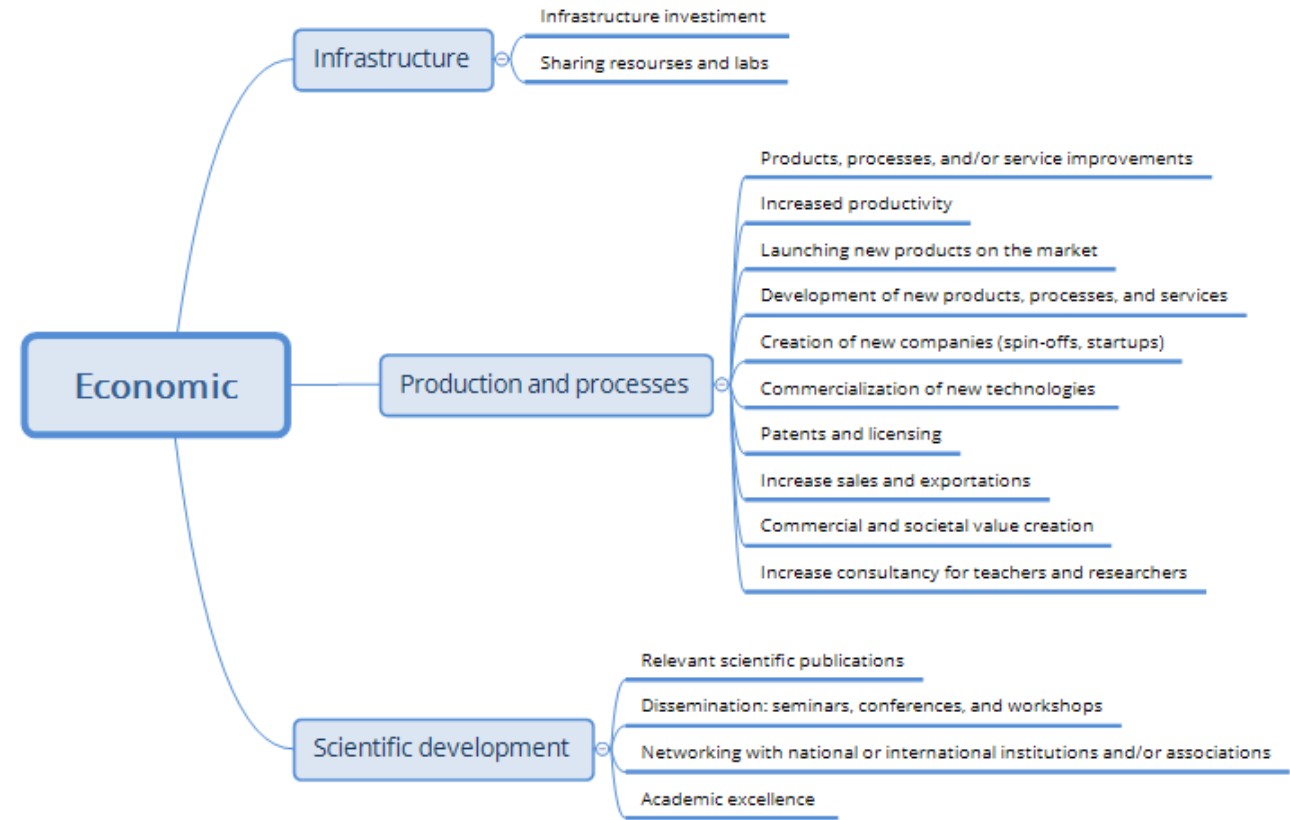

**Figure A1.** Economic impacts of university–industry collaborations.

**Table A4.** Economic impacts of university–industry collaborations.

| Economic Impact | Presences | Percentage (%) |
|---|---|---|
| Infrastructure investment | 2 | 2.13% |
| Sharing resources and labs | 9 | 9.57% |
| Product, process, and/or service improvements | 4 | 4.26% |
| Increased productivity | 8 | 8.51% |
| Launching new products on the market | 1 | 1.06% |
| Development of new products, processes, and services | 4 | 4.26% |
| Creation of new companies | 10 | 10.64% |
| Creation of spin-offs | 27 | 28.72% |
| Creation of startups | 13 | 13.83% |
| Commercialization of new technologies | 8 | 8.51% |
| Patents and licensing | 54 | 57.45% |
| Increased sales | 10 | 10.64% |
| Increased exportations | 2 | 2.13% |
| Commercial and societal value creation | 2 | 2.13% |
| Increased consultancy for professors and researchers | 17 | 18.09% |
| Relevant scientific publications | 23 | 24.47% |
| Dissemination: seminars, conferences, and workshops | 10 | 10.64% |
| Networking with national or international institutions and/or associations | 5 | 5.32% |
| Academic excellence | 2 | 2.13% |

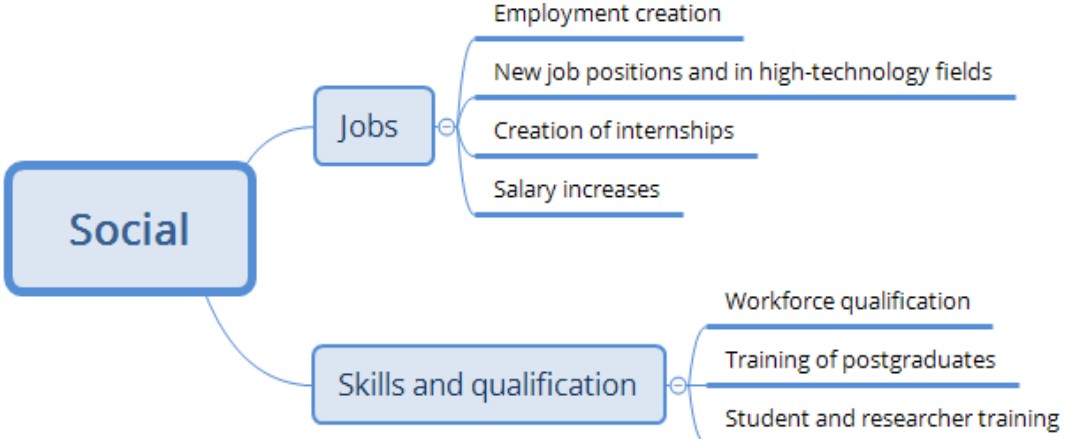

**Figure A2.** Social impacts of university–industry collaborations.

**Table A5.** Social impacts of university–industry collaborations.

| Social Impact | Presences | Percentage |
|---|---|---|
| Employment creation | 40 | 42.55% |
| New job positions and in high-technology fields | 7 | 7.45% |
| Creation of internships | 3 | 3.19% |
| Salary increases | 1 | 1.06% |
| Workforce qualification | 7 | 7.45% |
| Training of postgraduates | 1 | 1.06% |
| Student and researcher training | 4 | 4.26% |

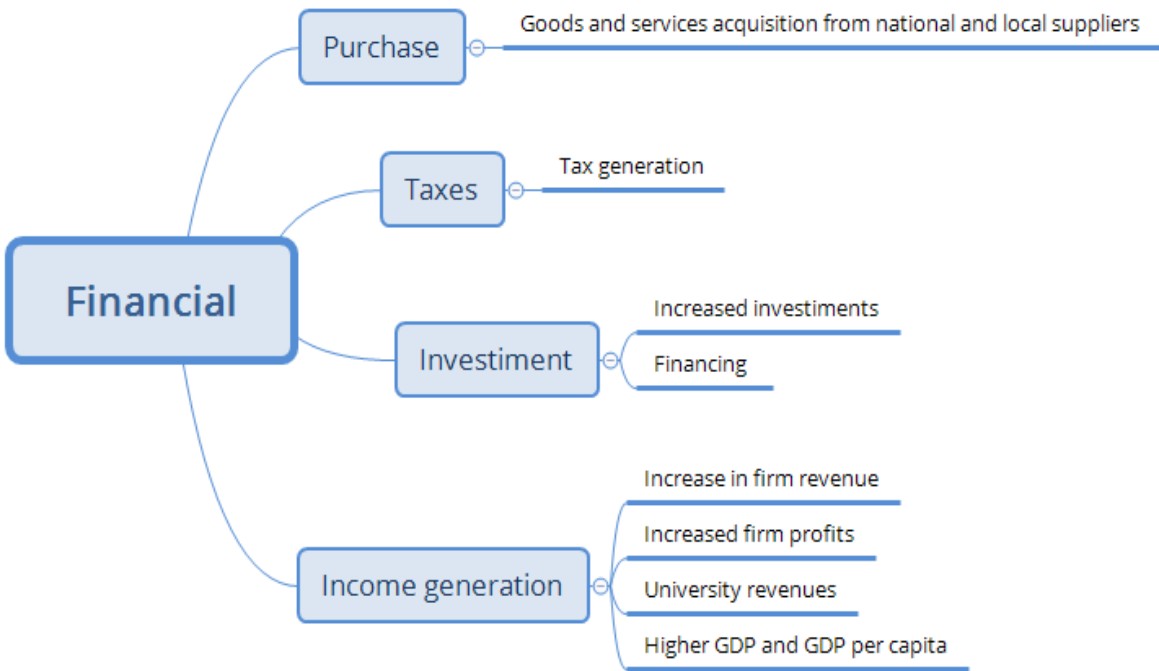

**Figure A3.** Financial impacts of university–industry collaborations.

**Table A6.** Financial impacts of university–industry collaborations.

| Financial Impact | Presences | Percentage |
|---|---|---|
| Good and services acquisition from national and local suppliers | 1 | 1.06% |
| Tax generation | 4 | 4.26% |
| Investments | 11 | 11.70% |
| Financing | 9 | 9.57% |
| Firm revenue | 1 | 1.06% |
| Firm profits | 7 | 7.45% |
| University revenues | 3 | 3.19% |
| Higher GDP and GDP per capita | 13 | 13.83% |

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
