# Peer review of "Socioeconomic Impacts of University–Industry Collaborations–A Systematic Review and Conceptual Model"

_2199-8531, doi:10.3390/joitmc7020137_

Round 1
Reviewer 1 Report
The article entitled “Socioeconomic Impacts of University – Industry Collaborations - A Systematic Review and Conceptual Model" is a timely and interesting research.
In my understanding, the methodological process is well conducted, although some parts can be better explained, namely: - a better explanation of the use of Scopus and WoS. In other words, there are some interesting scientific articles that announce the scientific areas where databases have the greatest potential. I, therefore, suggest that the use of those arguments; - a short explanation can be given on how the “content analysis” technique was carried out (i.e. my recommendation is to better explain each phase).
The article fails to the extent that it presents too many figures/tables and, in some cases, an in-depth discussion is not elaborated. My suggestion is to send part of these figures/tables to an appendix. Otherwise, the authors will have to describe the content of the figures in detail - for example, figure 9 seems to me that is not explained in detail.
For a better reading of the conclusions, I suggest dividing this chapter into subsections: - contributions to theory; - managerial contribution; - research limitations; - future research directions (section 4.1.5 should be sent to this chapter).
Overall, I think the authors present a good paper, but the organization of the article is still a little bit confusing. I think the article would benefit if it were cleaner, with fewer tables/figures in the main text and some information in one or several appendixes.
I would also recommend English proofreading before publication.
Congratulations for your work.
Reviewer 2 Report
I think this study is very necessary in that it delivers a comprehensive view of the socio-economic effects of industry-university cooperation.
- Although this study mainly evaluated the literature up to 2019, it would be good to add it to additionally review the following literature after 2020 and draw implications.
Malik, A., Sharma, P., Pereira, V., & Temouri, Y. (2021). From regional innovation systems to global innovation hubs: Evidence of a Quadruple Helix from an emerging economy. Journal of Business Research, 128, 587-598.
Bellandi, M., Donati, L., & Cataneo, A. (2021). Social innovation governance and the role of universities: Cases of quadruple helix partnerships in Italy. Technological Forecasting and Social Change, 164, 120518.
Hasche, N., Höglund, L., & Linton, G. (2020). Quadruple helix as a network of relationships: creating value within a Swedish regional innovation system. Journal of Small Business & Entrepreneurship, 32(6), 523-544.
Hou, B., Hong, J., & Shi, X. (2021). Efficiency of university–industry collaboration and its determinants: Evidence from Chinese leading universities. Industry and Innovation, 28(4), 456-485.
Sjöö, K., & Hellström, T. (2021). The two sides of the coin: joint project leader interaction in university‐industry collaboration projects. R&D Management.
Bastos, E. C., Sengik, A. R., & Tello-Gamarra, J. (2021). Fifty years of University-industry collaboration: a global bibliometrics overview. Science and Public Policy.
- In order to thoroughly review the literature related to this study, the following two things need to be confirmed, and these must be clearly described.
- First, were there any discussions or expert reviews about keywords?
- Second, did you go through the process of peer review in the process of removing documents?
- Tables 2, 3, and 4 may seem simpler if they are based on numbers. (number in table2, articles in table 3, participation in number of publications in table 4) And table 6 recommends moving to the appendix.
- Although this study described "CIMO Analysis" in section 4.1, detailed descriptions are needed to help readers understand more.
- Although this study focuses on university-industry cooperation, it is necessary to add what role the government from the triple helix perspective and the society from the quadruple helix perspective to further strengthen the socioeconomic impact.
- This study does not clearly describe the limitations of the study. We need to add this.

Reviewer 3 Report
I am impressed by the quantity of work which has been invested in this article: this is a very intensive review of literature about academy-industry collaboration, from a variety of perspectives.
I think the analysis is quite comprehensive and well organized. The anchor of the triple helix is well chosen and used. The quantity of technical details is sometimes even quite heavy (8 full pages!), the authors may consider some reductions.
The academic contribution is good but not exciting. There is no real analysis of the efficiency of different mechanisms or of the dependence on patterns of relations. I don't think that the authors can conclude that "university-industry collaborations have the potential to generate appropriate economic and social benefits". This is probably true, but not a result of a concrete analysis. The authors themselves propose such analyses to be done in the future.
I liked the figure 9 about the "socio-economic triple helix", and the relevant text. Still I think that one element of the outcomes should be mentioned: academy contributes mostly to knowledge creation, less to innovation concrete products or services.
Reviewer 4 Report
Authors must make the following corrections in the paper:
- Authors should explain better the academic contribution of the work developed.
Round 2
Reviewer 1 Report
The new version of the article is better, clearer and makes a stronger contribution.
Reviewer 2 Report
Overall, I think the authors did their best to do the review process. Nevertheless, the authors recommend supplementing/correcting the following two things.
The latest papers that I have recommended (papers since 2019) can enrich the discussion or be added to the limit/future research section of the paper.
And it seems inappropriate to describe the CIMO analysis method in 4.1. I recommend moving that part to 2.2.